# A Triazole Hybrid of Neolignans as a Potential Antileishmanial Agent by Triggering Mitochondrial Dysfunction

**DOI:** 10.3390/molecules25010037

**Published:** 2019-12-20

**Authors:** Carla Cardozo Pinto de Arruda, Daiana de Jesus Hardoim, Yasmin Silva Rizk, Celeste da Silva Freitas de Souza, Tânia Zaverucha do Valle, Diego Bento Carvalho, Noemi Nosomi Taniwaki, Adriano Cesar de Morais Baroni, Kátia da Silva Calabrese

**Affiliations:** 1Laboratório de Parasitologia Humana, Instituto de Biociências, Universidade Federal de Mato Grosso do Sul, 79070-900 Campo Grande, Mato Grosso do Sul, Brasil; 2Laboratório de Imunomodulação e Protozoologia (LIMP), Instituto Oswaldo Cruz (IOC), Fundação Oswaldo Cruz (FIOCRUZ), 21040-360 Rio de Janeiro, Rio de Janeiro, Brasil; daianahardoim@gmail.com (D.d.J.H.); yasminrizk@gmail.com (Y.S.R.); celcroix@gmail.com (C.d.S.F.d.S.); taniazv@gmail.com (T.Z.d.V.); 3Laboratório de Síntese e Química Medicinal (LASQUIM), Faculdade de Ciências Farmacêuticas, Alimentos e Nutrição, Universidade Federal de Mato Grosso do Sul, 79070-900 Campo Grande, Mato Grosso do Sul, Brasil; diegob.carvolho@hotmail.com (D.B.C.); adrianobaroni@hotmail.com (A.C.d.M.B.); 4Núcleo de Microscopia Eletrônica, Instituto Adolfo Lutz, 01246-000 São Paulo, São Paulo, Brasil; ntaniwak@hotmail.com

**Keywords:** *Leishmania amazonensis*, synthetic derivative, antileishmanial activity, cell death, ultrastructure

## Abstract

In the search for new compounds with antileishmanial activity, we synthesized a triazole hybrid analogue of the neolignans grandisin and machilin G (LASQUIM 25), which was previously found highly active against both promastigotes and intracellular amastigote forms of *Leishmania amazonensis*. In this work, we investigated the leishmanicidal effects of LASQUIM 25 to identify the mechanisms involved in the cell death of *L. amazonensis* promastigotes. Transmission electron microscopy (TEM) analysis showed marked effects of LASQUIM 25 (IC_50_ = 7.2 µM) on the morphology of promastigote forms, notably on mitochondria. The direct action of the triazole derivative on the parasite was noticed over time from 2 h to 48 h, and cells displayed several ultrastructural alterations characteristic of apoptotic cells. Also, flow cytometric analysis (FACS) after TMRE staining detected changes in mitochondrial membrane potential after LASQUIM 25 treatment (64.83% labeling versus 83.38% labeling in nontreated cells). On the other hand, FACS after PI staining in 24 h-treatment showed a slight alteration in the integrity of the cell membrane, a necrotic event (16.76% necrotic cells versus 3.19% staining in live parasites). An abnormal secretion of lipids was observed, suggesting an exocytic activity. Another striking finding was the presence of autophagy-related lysosome-like vacuoles, suggesting an autophagic cell death that may arise as consequence of mitochondrial stress. Taken together, these results suggest that LASQUIM 25 leishmanicidal mechanisms involve some degree of mitochondrial dysregulation, already evidenced by the treatment with the IC_50_ of this compound. This effect may be due to the presence of a methylenedioxy group originated from machilin G, whose toxicity has been associated with the capacity to generate electrophilic intermediates.

## 1. Introduction

Leishmaniases are vector-borne parasitic diseases included in the neglected tropical diseases (NTDs) group according to the World Health Organization [1]. The great variety of causative species belonging to the *Leishmania* genus determines a wild spectrum of clinical manifestations, mainly the tegumentary and the visceral forms of the disease.

The treatment recommended by WHO is based on the use of pentavalent antimonials, having amphotericin B, miltefosine, and paromomycin as therapeutic options [1]. All these drugs have many issues related to toxicity, cost, and resistance by the parasite. Therefore, it becomes imperative the search for new antileishmanials, which could bring a new breath for patients and physicians. 

In the search for new compounds with antileishmanial activity, our research group has synthesized a class of molecules derived from neolignans veraguensin **1**, grandisin **2**, and machilin G **3**, through bioisosteric replacement of the tetrahydrofuran ring by the 1,2,3-triazole core (Figure 1) [2]. Neolignans are privileged structures in medicinal chemistry since they have different biological activities, among them the antileishmanial [3,4,5,6]. Among 16 synthesized compounds, the triazole hybrid **6** (**LASQUIM 25**) derived from grandisin **2** and machilin G **3** was highly active against both the flagellated and the intracellular target forms of *Leishmania (Leishmania) amazonensis* [2,3]. This species is the causative agent of diffuse cutaneous leishmaniasis cases in the New World, which have difficult clinical management [7].

Here we investigated the leishmanicidal effects of LASQUIM 25 **6** to identify the mechanisms involved in the cell death of *L. amazonensis* promastigotes. The results suggest a direct action of LASQUIM 25 **6** on the parasites, with ultrastructural alterations characteristic of apoptosis and autophagic cell death, which may arise as a consequence of mitochondrial stress.

## 2. Results and Discussion

### 2.1. Transmission Electron Microscopy (TEM)

TEM was performed to evaluate ultrastructural alterations caused by LASQUIM 25 **6** on *L. amazonensis* promastigote forms treated with the half-maximal inhibitory concentration (IC_50_) calculated for the triazole derivative (7.2 µM) [2]. Figure 2A,B shows the well-preserved cell morphology of nontreated parasites, with characteristic elongated fusiform shape, and a rounded nucleus with peripherally condensed chromatin. Both the plasma membrane and the flagellar pocket were intact but some lipid bodies were observed in the cytoplasm. Figure 2C shows promastigotes treated for 2 h with the IC_50_ of triazole derivative **6**. Clear vacuoles with electron-dense borders suggestive of acidocalcisomes were observed in the cytoplasm [8]. After 4 h-treatment greater electron-dense inclusions were also observed (Figure 2D).

After 8 h-treatment, cytoplasmic vesicles were noticed. Besides, the drug-induced vesicles formation, observed in the flagellar pocket (Figure 3A). These findings suggest an exocytic activity as a result of an abnormal secretion of lipids or an exacerbated production of proteins in an attempt of the parasite to survive [9,10]. Figure 3B shows swollen mitochondria observed after 16 h-treatment. This finding was more pronounced after 24 h when the loss of cell shape was also observed (Figure 3C). Another striking finding is the presence of autophagy-related lysosome-like vacuoles, which are probably associated with drug-induced cellular stress. Finally, after 48 h-treatment, swelling of mitochondria was even more pronounced (Figure 3D). It was equally observed numerous vesicles with granular material throughout the cytoplasm. Promastigote forms seem to lose their cellular integrity, with severe cellular damage expressed by internal disorganization and fragmentation of organelles.

Thus, the direct action of the triazole derivative **6** on the parasite was evidenced over time, and cells displayed several ultrastructural alterations characteristic of apoptotic cells. Promastigote cytoplasm seemed less electron-dense, with intense cytoplasmic vacuolization. LASQUIM 25 **6** also induced cell shrinkage and a rounded appearance of promastigotes. This probably occurred due to the loss of cytoskeletal integrity triggered by apoptotic signals. Similar processes of cytoskeletal disorganization were described in the literature [11,12]. Despite this, the cell membrane showed no ruptures. Necrosis is induced by a breakdown of the intracellular membrane, causing the release of intracellular constituents, which was not observed in our experiments.

The mitochondrial injury was expected since a depolarization of the mitochondrial membrane potential was noticed following TMRE assay, described in the next section. Thus, LASQUIM 25 **6** may have triggered the oxidative stress related to the mitochondrial dysregulation, an overriding feature of apoptosis [13]. Several studies show the oxidative stress and consequent mitochondrial dysfunction as an effect of methylenedioxy-containing compounds [14,15].

Toxicity of the methylenedioxy group has been associated with the capacity to generate electrophilic intermediates [16]. Methylenedioxy-containing compounds may exert inhibitory effects on cytochrome P450 enzymes (CYP450s) [17]. CYP450s are hemoproteins that catalyze many important biochemical reactions, as in the metabolism of xenobiotics and synthesis of sterols. They are functionally important for the survival of invasive pathogens such as *Leishmania* sp.

Another hypothesis to consider is the autophagic cell death, due to the observation of autophagy-related lysosome-like vacuoles after a 24 h-treatment with LASQUIM 25. In autophagy, subcellular membranes enclosing cytoplasm and organelles transport them to lysosomes or vacuoles for degradation, leading to a vacuolization of the cytoplasm without condensation of chromatin [18]. Some authors have postulated that the induction of *Leishmania* cell death may be triggered by the rupture of membranes of lysosomes/lysosome-like vacuoles and the release of several digestive lysosomal enzymes into the cytoplasm, leading to complete intracellular digestion of parasites surrounded by an intact cellular membrane [19,20]. This unique cell death mechanism may have developed in these parasites to avoid the induction of an antileishmanial immune response in the host [19].

### 2.2. Flow Cytometric Analysis (FACS)

#### 2.2.1. PI Staining

Necrosis is detected by measuring the plasma membrane permeability to propidium iodide (PI), a DNA-binding, normally impermeable fluorescent dye. LASQUIM 25 **6** (7.2 µM) seems to induce a slight alteration in the promastigote cell membrane integrity. PI-labeled treated promastigotes did not show significant permeabilization of the plasma membrane compared with nontreated parasites, which is in line with the ultrastructural observations. PI labeling after 24 h-treatment with IC_50_ of LASQUIM 25 **6** showed 16.76% necrotic cells (Figure 4C), slightly higher than the negative control with live parasites (3.19%, Figure 4A). After 48 h-treatment, labeling practically doubled, reaching 30.34% (Figure 5C), but the marking practically increased six-fold in the negative control (18.46%). Heat-killed parasites showed 96.36% and 94.56% labeling, respectively after 24 h- and 48 h-treatment.

The reference drug Amphotericin B (0.15 μM) induced greater alteration in cell membrane integrity (Figure 4D and Figure 5D). Treated parasites showed 57.03% and 63.40% staining, respectively after 24 h (Figure 4D) and 48 h-treatment (Figure 5D).

#### 2.2.2. TMRE Staining

Tetramethylrhodamine, ethyl ester (TMRE) is a cell-permeant, fluorescent dye that is readily sequestered by active mitochondria. It was observed changes in mitochondrial membrane potential after LASQUIM 25 **6** treatment (7.2 µM). TMRE labeling after 24h-treatment was 64.83% (Figure 6C) when nontreated cells’ labeling was 83.38% (Figure 6A). After 48h-treatment (Figure 7), the effect of the triazole derivative was very similar (61.35% active mitochondria). During apoptosis, mitochondrial potential reduction precedes most of the morphological changes occurring during the apoptotic process and before phosphatidylserine (PS) exposure in the outer leaflet of the plasma membrane [21,22]. Thus, we believe that apoptosis may be one of the processes associated with the cell death of the treated promastigotes. The alterations after TMRE labeling are in agreement with the ultrastructural analysis, where the kinetoplast-mitochondrial complex swelling was observed.

The reference drug amphotericin B strongly interfered with the mitochondrial activity (Figure 6D and Figure 7D). After 24h-treatment, TMRE labeling was quite similar to the positive control, heat-killed parasites (6.62% versus 5.0%; Figure 6B,D).

The apoptotic pathway in *Leishmania* is not clear, once the classical mammalian key proteins of apoptosis have not been described in this parasite [23]. Basmaciyan et al. [23] observed that amphotericin B induced apoptosis in *L. major* parasites by the inhibition of the metacaspase LmjMCA, a cysteine peptidase that shares similarities with caspases but has different substrate specificity. The authors postulated that LmjMCA inhibition directly induced apoptosis, which could be explained by the role of LmjMCA in the autophagic cell survival process [24] and the fact that inhibiting autophagy induces cell apoptosis under stress [25]. Another hypothesis was that amphotericin B induced apoptosis by an LmjMCA-independent pathway.

Taken together, FACS and ultrastructural analysis suggest LASQUIM 25 **6** leishmanicidal mechanisms involve some degree of mitochondrial dysregulation, as evidenced by the treatment with the IC_50_ value. Apoptosis and necrosis induce the transition of mitochondrial membrane permeability (MPT), which results from the opening of a high conduction pore on the inner mitochondrial membrane. Mitochondria is a potential target for antileishmanial drugs and this mechanism of action has been widely described [11,12,26,27]. The single mitochondria of the kinetoplastid parasite and its functional features are markedly distinct from the mammalian ones [28]. Furthermore, apoptotic death encompasses the change in the cellular form, notably shrinkage and reduction of cell size, as described here as a LASQUIM 25 **6** effect.

Nevertheless, the hypothesis of an autophagy cannot be discarded. In this process, characteristic autophagic vacuoles encompass lipids that were not incorporated into the cellular membranes [26]. Interestingly, autophagic cell death may arise as a consequence of mitochondrial stress [29].

## 3. Materials and Methods

### 3.1. Chemistry

Terminal acetylene **7**, aryl azide **8**, and triazole in Appendix A analogue LASQUIM 25 **6** were synthesized using procedures from our research group [2], as seen in Appendix A (Figure 1, Reference [2]).

### 3.2. Parasites

The standard strain IFLA/BR/1967/PH8 of *L. amazonensis* was used in in vitro antileishmanial tests. Amastigotes were routinely isolated from BALB/c mice cutaneous lesions and maintained as promastigotes from up to ten serial passages at 26 °C in Schneider’s Insect Medium (Sigma-Aldrich, St. Louis, MO, USA) supplemented with 20% of fetal calf serum (FCS, Sigma-Aldrich), 10,000 U/mL penicillin, and 10 mg/mL streptomycin (Sigma-Aldrich).

### 3.3. In Vitro Tests

#### 3.3.1. Transmission Electron Microscopy

Transmission electron microscopy (TEM) was performed on *L. amazonensis* promastigote forms treated with the half-maximal inhibitory concentration (IC_50_) calculated for the triazole derivative LASQUIM 25 **6** (7.2 µM) [2]. Nontreated parasites were used as negative control. After 2, 4, 8, 16, 24, and 48 h-incubation promastigotes were collected by centrifugation at 1500× *g*. Cells were washed, fixed, and embedded in Epon^®^ resin (Polysciences, Inc., Warrington, PA, USA) [30]. Ultrathin sections were stained with uranyl acetate and lead citrate. Parasites were examined in a Jeol 1011 transmission electron microscopy (Jeol, Inc., Peabody, MA, USA).

#### 3.3.2. Flow Cytometric Analysis for Detection of Cellular Membrane Alterations

*L. amazonensis* promastigote forms (1 × 10^6^/mL) were treated for 24 and 48 h with calculated IC_50_ (7.2 µM) [2], in Schneider’s Insect Medium (Sigma-Aldrich) supplemented with 20% FCS (Cultilab, Campinas, SP, Brazil), 10,000 U/mL penicillin, and 10 mg/mL streptomycin (Sigma-Aldrich). Heat-killed parasites (60 °C bath/10 min) were used as positive control and nontreated parasites were used as negative control. Amphotericin B was used as the reference drug at calculated IC_50_ (0.15 µM) [2]. Promastigotes were stained with 10 µM propidium iodide (BD Pharmingen^TM^) and immediately submitted to flow cytometric analysis (BD FACScalibur^TM^, Aalst, East Flanders, Belgium). A total of 10,000 events were acquired in the region previously established as corresponding to the parasites. Fluorescence was quantified and histograms were designed [30].

#### 3.3.3. Flow Cytometric Analysis for Detection of Mitochondrial Membrane Potential (ΔΨ_m_) Alterations

*L. amazonensis* promastigote forms (1 × 10^6^/mL) were treated for 24 and 48 h with calculated IC_50_ (7.2 µM) [2], in Schneider’s Insect Medium (Sigma-Aldrich) supplemented with 20% FCS (Cultilab^®^), 10,000 U/mL penicillin, and 10 mg/mL streptomycin (Sigma-Aldrich). Heat-killed parasites (60 °C bath / 10 min) were used as positive control and nontreated parasites were used as negative control. Amphotericin B was used as the reference drug at calculated IC_50_ (0.15 µM) [2]. Promastigotes were stained with 50 nM Tetramethylrhodamine, Ethyl Ester (TMRE) (Molecular Probes, Carlsbad, CA, USA) for 15 min at room temperature, and submitted to flow cytometric analysis (BD FACScalibur^TM^). A total of 10,000 events were acquired in the region previously established as corresponding to the parasites. Fluorescence was quantified and histograms were designed [30].

## Figures and Tables

**Figure 1 molecules-25-00037-f001:**
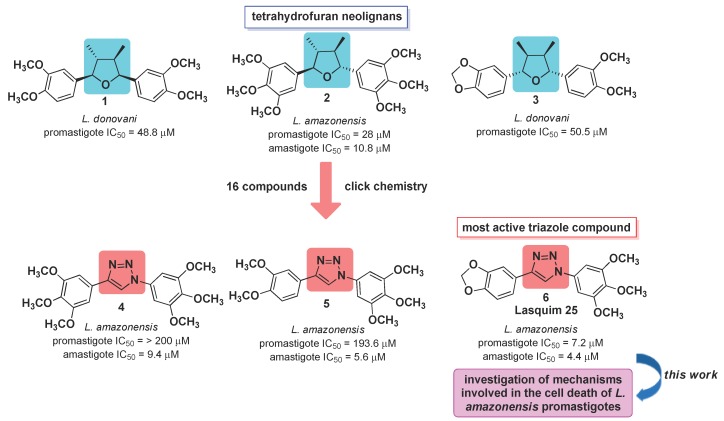
Antileishmanial activities of synthetic triazole compounds **4–6** based on tetrahydrofuran neolignans **1–3**.

**Figure 2 molecules-25-00037-f002:**
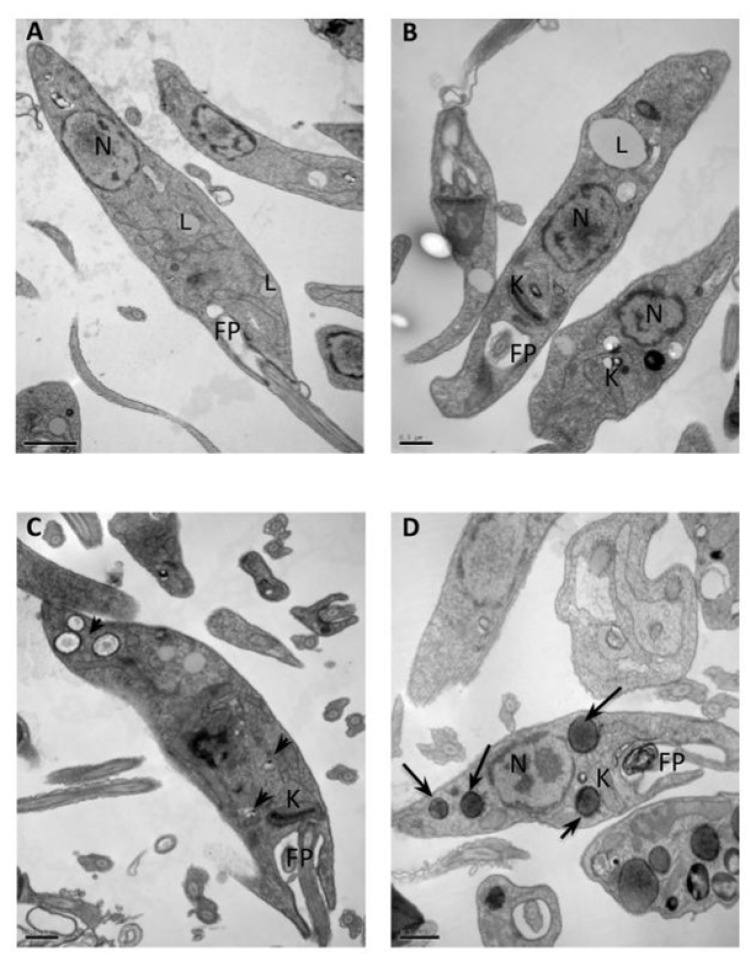
Transmission electron microscopy of *L. amazonensis* promastigotes treated with triazole derivative of neolignans LASQUIM 25 (7.2 μM). (**A**,**B**) Nontreated parasites showing characteristic morphology of kinetoplast (K), flagellar pocket (FP), and nucleus (N). Lipid droplets were observed (L). (**C**) Presence of alcidocalcisomes (arrowheads) after 2 h-treatment with the synthetic compound. (**D**) Electron-dense inclusions were observed after 4 h-treatment (arrows).

**Figure 3 molecules-25-00037-f003:**
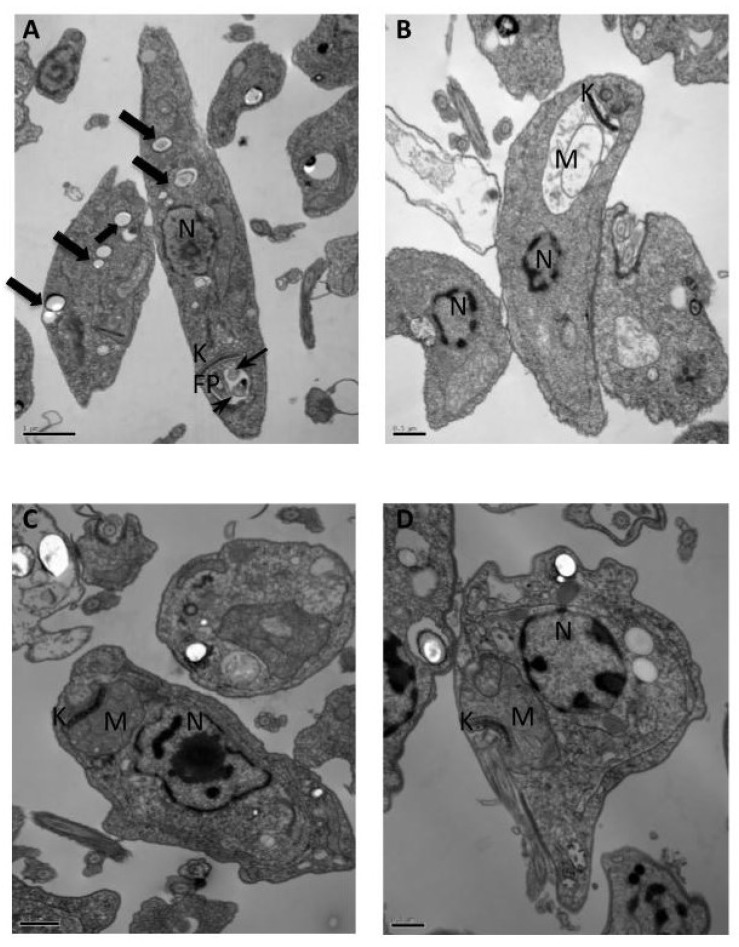
Ultrastructural effects of triazole derivative LASQUIM 25 (7.2 μM) on *L. amazonensis* promastigotes. (**A**) 8 h-treatment showing cytoplasmic vesicles (large arrows) and inclusions in the flagellar pocket (arrows). (**B**) Swollen mitochondria (M) observed after 16 h-treatment. (**C**) Swollen mitochondria (M), granular cytoplasmic inclusions, and loss of cellular shape after 24 h-treatment. (**D**) 48 h-treatment showing cellular disorganization and severe damage to the cytoplasm, with numerous vesicles with granular material; swollen mitochondria (M).

**Figure 4 molecules-25-00037-f004:**
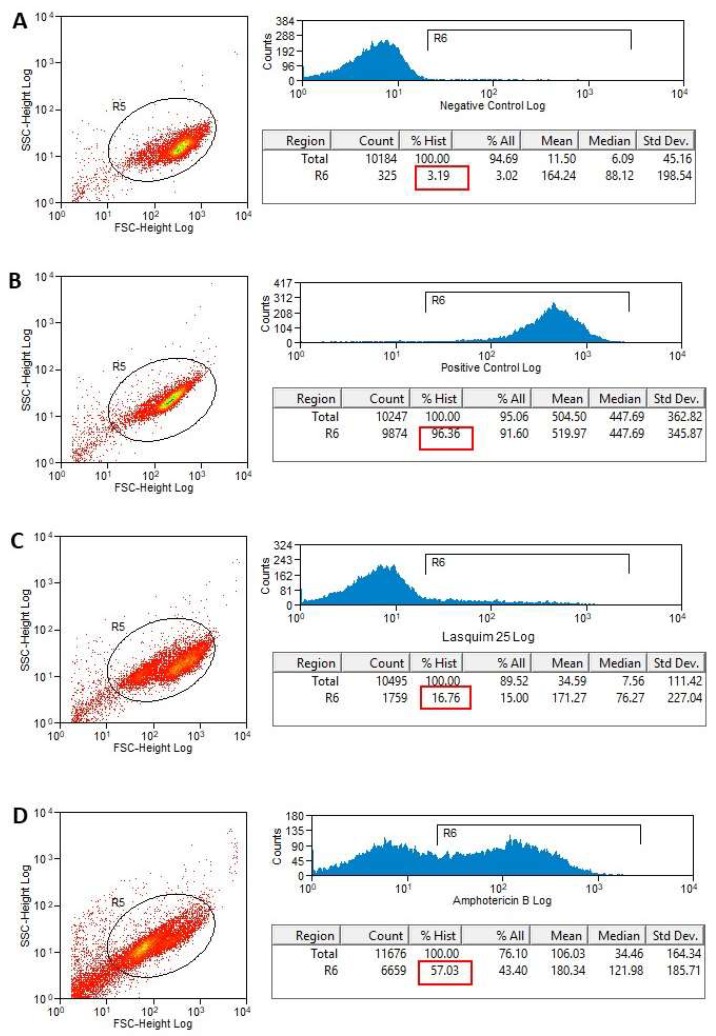
Flow cytometry of *L. amazonensis* to evaluate the plasma membrane permeability to propidium iodide (PI) after 24 h-treatment. Promastigotes captured in the gated region and representative histograms. (**A**) Nontreated promastigotes (Negative Control). (**B**) Heat-killed parasites (Positive Control). (**C**) Promastigotes treated with LASQUIM 25 (7.2 μM). (**D**) Promastigotes treated with the reference drug Amphotericin B (0.15 μM).

**Figure 5 molecules-25-00037-f005:**
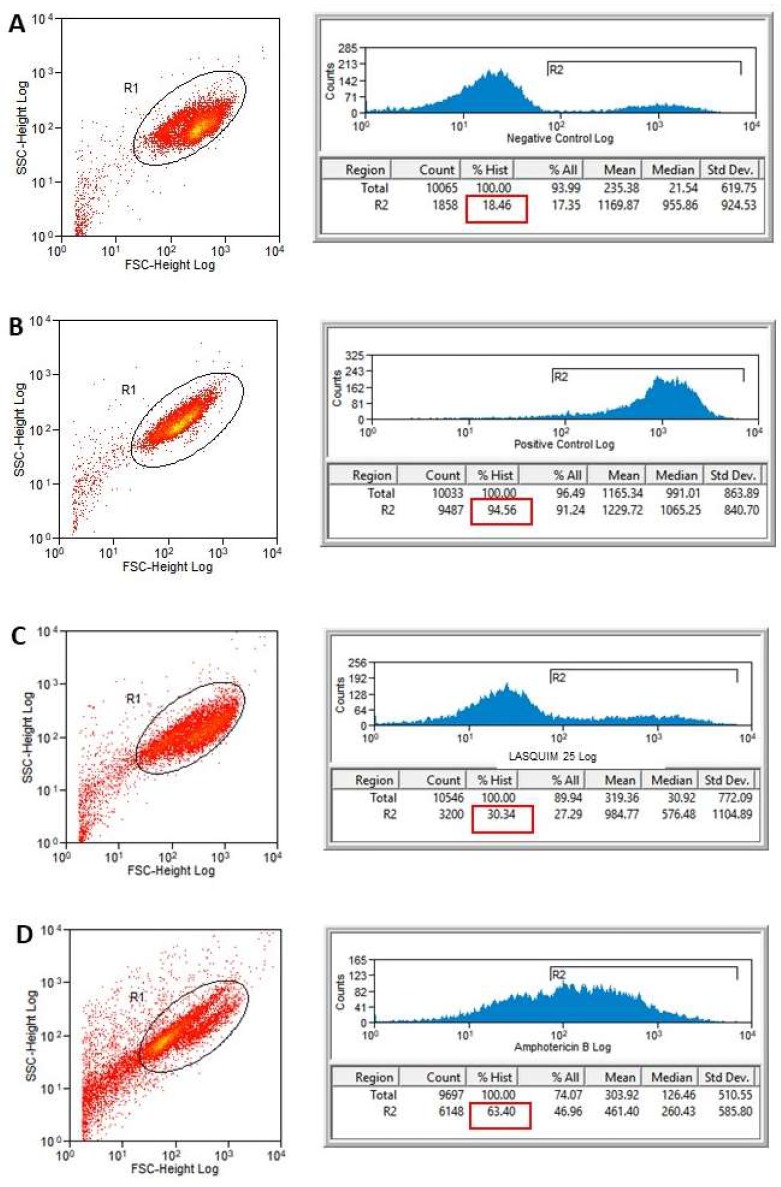
Flow cytometry of *L. amazonensis* to evaluate the plasma membrane permeability to propidium iodide (PI) after 48 h-treatment. Promastigotes captured in the gated region and representative histograms. (**A**) Nontreated promastigotes (Negative Control). (**B**) Heat-killed parasites (Positive Control). (**C**) Promastigotes treated with LASQUIM 25 (7.2 μM). (**D**) Promastigotes treated with the reference drug Amphotericin B (0.15 μM).

**Figure 6 molecules-25-00037-f006:**
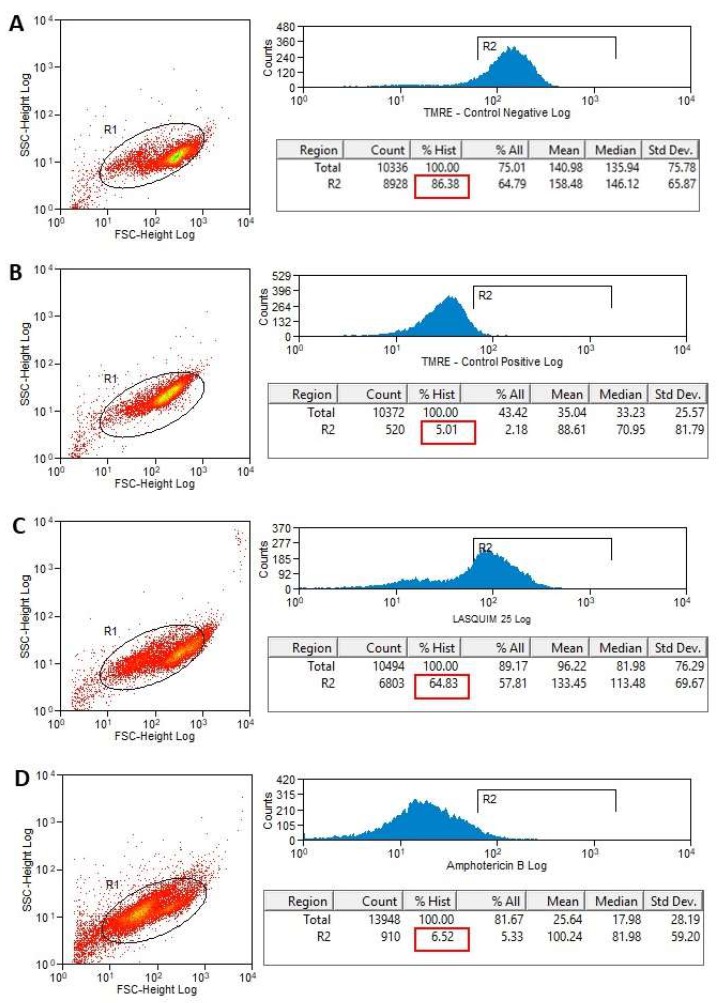
Flow cytometry of *L. amazonensis* to evaluate the mitochondrial membrane potential (ΔΨ_m_) after 24 h-treatment. Promastigotes captured in the gated region and representative histograms of promastigotes incubated with TMRE. (**A**) Nontreated promastigotes (Negative Control). (**B**) Heat-killed parasites (Positive Control). (**C**) Promastigotes treated with LASQUIM 25 (7.2 μM). (**D**) Promastigotes treated with the reference drug Amphotericin B (0.15 μM).

**Figure 7 molecules-25-00037-f007:**
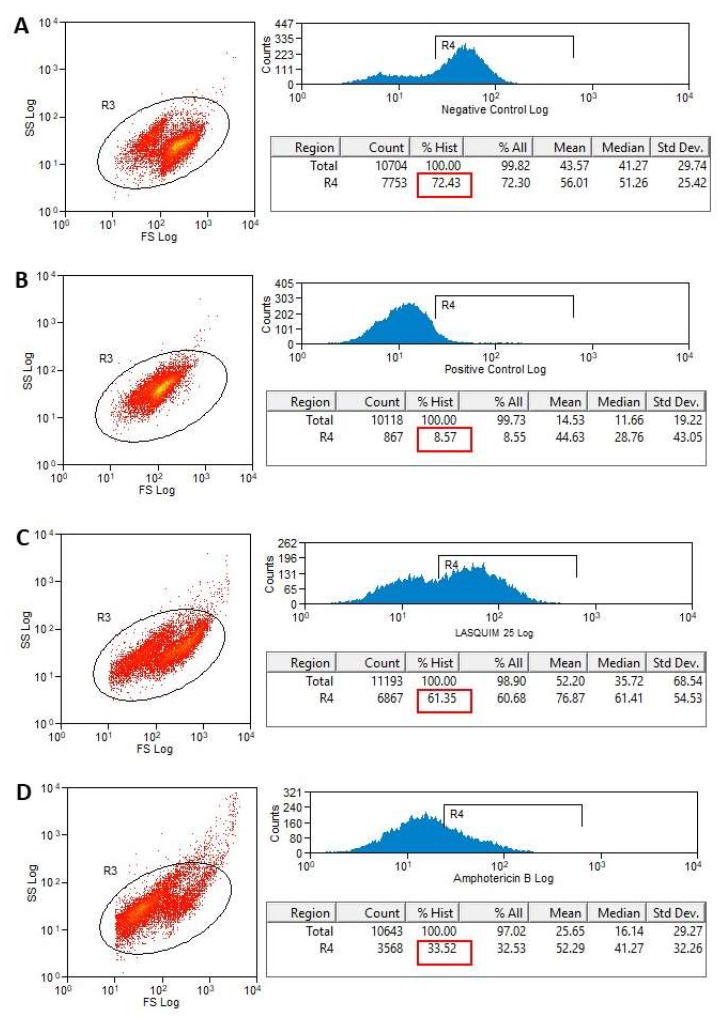
Flow cytometry of *L. amazonensis* to evaluate the mitochondrial membrane potential (ΔΨ_m_) after 48 h-treatment. Promastigotes captured in the gated region and representative histograms of promastigotes incubated with TMRE. (**A**) Nontreated promastigotes (Negative Control). (**B**) Heat-killed parasites (Positive Control). (**C**) Promastigotes treated with LASQUIM 25 (7.2 μM). (**D**) Promastigotes treated with the reference drug Amphotericin B (0.15 μM).

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
