# Peer review of "A Triazole Hybrid of Neolignans as a Potential Antileishmanial Agent by Triggering Mitochondrial Dysfunction"

_molecules, 2019, doi:10.3390/molecules25010037_

Round 1

Reviewer 1 Report

Manuscript “A triazole hybrid of neolignans as a potential antileishmanial agent by triggering mitochondrial dysfunction” does not correspond to the section - Medicinal chemistry, because it does not contain any chemical research. It only describes biological studies on one chemical compound obtained previously - citation 28. No synthesis scheme is needed (page 11). The synthesis method should also be removed - verses 228 - 235. The manuscript is correctly prepared but should be moved to another section.

Author Response

Response to Reviewer 1 Comments

We are pleased to submit to your evaluation a revised version of our manuscript. We would like to thank you for the valuable comments.

Point 1: Manuscript “A triazole hybrid of neolignans as a potential antileishmanial agent by triggering mitochondrial dysfunction” does not correspond to the section - Medicinal chemistry, because it does not contain any chemical research. It only describes biological studies on one chemical compound obtained previously - citation 28. The manuscript is correctly prepared but should be moved to another section.

Response 1: According to the journal Molecules, the Medicinal Chemistry Section encompasses original research and review articles that increase the understanding of how the chemical structure of bioactive molecules determines their pharmacodynamic, pharmacokinetic, and physicochemical properties, and, hence their therapeutic potential. In particular, this Section mainly invites contributions that report on: the design, synthesis, chemical characterization, and biological evaluation of novel compounds against biological targets of therapeutic, diagnostic, or theranostic interest, and/or the assessment of their physicochemical and pharmacokinetic properties.

So, we believe our manuscript fits in this Section, as we have described the leishmanicidal effects of a synthetic compound.

Also, we have previously submitted the abstract of the manuscript in advance for an Assistant Editor of Molecules, and she suggested the mentioned section.

Point 2: No synthesis scheme is needed (page 11). The synthesis method should also be removed - verses 228 - 235.

Response 2: The synthesis scheme was moved to Supporting Information and briefly explained according to the references of origin (citations 2 and 3).

Reviewer 2 Report

The manuscript molecules-594807 needs a file of supporting information. This file should include all NMR along with other spectroscopic details.

Author Response

We are pleased to submit to your evaluation a revised version of our manuscript. We would like to thank you for the valuable comments.

Point 1: The manuscript molecules-594807 needs a file of supporting information. This file should include all NMR along with other spectroscopic details.

Response 1: The synthesis scheme was moved to Supporting Information and briefly explained according to the references of origin (citations 2 and 3) containing NMR and spectroscopic details.

Reviewer 3 Report

This paper describes a triazole derivative with in vitro anti-leishmanial activity and proposes that the mechanism of action likely involves mitochondrial dysfunction based on microscopy and flow cytometry data. In general, the microscopy and flow cytometry experiments presented are well done and this is a well-written paper. In addition, reporting a new compound with anti-leishmanial activity will be of interest to researches in this particular field of microbiology. However, several aspects of the paper, including the lack of more in-depth mechanism studies, will likely reduce the overall impact of the work. In addition, several other issues with the paper reduced enthusiasm. In conclusion, I recommend this paper for publication in Molecules after major revisions.

The authors should address the following:

The structures of grandisin and machilin G should be included. It is unclear if grandisin and/or machilin G have anti-leishmanial activity, which formed the basis for the claim that the triazole is somehow a hybrid. A hybrid of what? A rationale for the specific selection of a triazole as a replacement of the tetrahydrofuran present in grandisin and machilin G should be included. In particular, the electronic and conformational differences should be discussed. Why not other 5-member heteroaromatics? Effects of the compound on mammalian cells and mitochondrial function should be included as a means to assess selectivity. The claim that the methylenedioxy is contributing to activity should be tested by comparing a derivative where this group is replaced with two methoxys that presents similar substituents as found in grandisin (which lacks a methylenedioxy). Redundancies in the references should be removed (e.g. reference 2 and 3 are the same as 28 and 29; reference 8 is the same as 23).

Author Response

Response to Reviewer 3 Comments

We are pleased to submit to your evaluation a revised version of our manuscript. We would like to thank you for the valuable comments.

Point 1: The structures of grandisin and machilin G should be included. It is unclear if grandisin and/or machilin G have anti-leishmanial activity, which formed the basis for the claim that the triazole is somehow a hybrid. A hybrid of what? A rationale for the specific selection of a triazole as a replacement of the tetrahydrofuran present in grandisin and machilin G should be included. In particular, the electronic and conformational differences should be discussed. Why not other 5-member heteroaromatics?

Response 1: We have reviewed the introduction. The structures of the neolignans veraguensin 1, grandisin 2 and machilin G 3 were included and their antileishmanial activities were cited as well (Silva Filho et al.  Phytother. Res.  2008, 22, 1307-1310; Neves et al. Chem. Biol. Drug Des. 2019, 00, 1-9).

Compounds were designed by the substitution of the tetrahydrofuran ring by the triazole ring (bioisosterism approach), generating 16 compounds. The triazole analogue 6, a hybrid of the neolignans 2 and 3, was the most active among the 16 compounds tested on both promastigote and amastigote forms of Leishmania amazonensis. Then it was selected in order to investigate its mechanism of action. Other heterocycles as substituents of the triazole core, such as the isoxazole analogues, were recently tested by our research group on L. amazonensis and the results were published (Trefzger et al. Chem Biol Drug Des. 2019, 93, 313–324; Neves et al, Chem. Biol. Drug Des. 2019, https://doi.org/10.1111/cbdd.13609). The mechanistic study of the isoxazole series is currently underway.

Point 2: Effects of the compound on mammalian cells and mitochondrial function should be included as a means to assess selectivity. The claim that the methylenedioxy is contributing to activity should be tested by comparing a derivative where this group is replaced with two methoxys that presents similar substituents as found in grandisin (which lacks a methylenedioxy).

Response 2: The selectivity was assessed previously by testing the cytotoxicity of the compound on J774.A1 cells (Selectivity Index (SI)=10.6, Costa et al., Molecules 2016, 21, 802) and NIH/3T3 fibroblasts (SI=12.5, Cassamale et al. J. Braz. Chem. Soc. 2016, 27, 1217-1228). We have previously tested the antileishmanial activity of the derivative with two methoxys (Costa et al., Molecules 2016, 21, 802; Cassamale et al. J. Braz. Chem. Soc. 2016, 27, 1217-1228). As we mentioned above, the triazole analogue 6 was the most active and it was selected for a qualitative study of mechanistic action. Studies on the leishmanicidal mechanism of the other compounds are being planned and will be published as soon as possible.

Point 3: Redundancies in the references should be removed (e.g. reference 2 and 3 are the same as 28 and 29; reference 8 is the same as 23). 

Response 3: Redundancies in the references were removed.

Round 2

Reviewer 1 Report

The manuscript can be accented in present form.

Reviewer 3 Report

The authors addressed the issues of the previous review.  I recommend this paper for Molecules.